# OpenReview forum: "TAIA: Large Language Models are Out-of-Distribution Data Learners"
_NeurIPS.cc/2024/Conference — NeurIPS 2024 poster_

### Official Review · Reviewer_VBCP · 2024-06-13

**Soundness:** 3
**Presentation:** 2
**Contribution:** 3
**Rating:** 6
**Confidence:** 3

**Summary:**

The paper introduces TAIA, a novel method for enhancing the performance of large language models (LLMs) in data-scarce domains with domain-mismatched data. The authors identify that during fine-tuning, only attention parameters significantly contribute to downstream task performance when training and test sets' distributions do not align. The proposed TAIA method retains updates to attention parameters while discarding updates to feed-forward network parameters, effectively improving performance on out-of-distribution (OOD) tasks. Extensive experiments demonstrate TAIA's superior performance across various datasets and model configurations, showcasing its robustness and generalizability.

**Strengths:**

- The TAIA method is novel to selectively retain beneficial parameter updates during fine-tuning.
- The paper provides a thorough analysis of the roles of self-attention and feed-forward networks in LLMs.
- The comprehensive experiments validate the proposed method.

**Weaknesses:**

- The presentation could be clear. For example, figure 2 could be more clear with a better caption. Currently, the caption is not informative for me to understand the method and its effectiveness. In addition, Figure 3 also made me confused. It seems that the base model performance pretty well. If this is the case, why should we use other methods?

**Questions:**

See weakness

**Limitations:**

yes

---

> ### Author Rebuttal · Authors · 2024-07-31
>
> Thank you for your review and important questions! We are glad to hear that you find our newly proposed TAIA useful novel and that our analysis on self-attention and feed-forward networks is thorough. Below we address the questions and concerns raised in the review.
>
> > For example, figure 2 could be more clear with a better caption. Currently, the caption is not informative for me to understand the method and its effectiveness.
>
> Response: Sorry for the confusion. Figure 2 compares the scenario when we train in-domain data (colored as green) and out-of-domain data (colored as yellow) and evaluate in in-domain test sets and out-of-domain test sets, respectively (The second row; fine-tuning). The vanilla fine-tuning method can only perform well when trained on ID data and evaluated in ID test sets. Compared to vanilla tuning, TAIA can perform generally well on both types of test sets when given OOD data. As a similar approach that only trains attention, TOA (Train-only-attention) performs badly on both types of evaluation sets as it loses sufficient exploration of optimal parameter groups. We will make the caption clear in the revised version.
>
> > In addition, Figure 3 also made me confused. It seems that the base model performance pretty well. If this is the case, why should we use other methods?
>
> Response:
> - Reasons of strong base models: We conduct experiments on out-of-domain data, which means that the base model never suffers from the catastrophic forgetting problem that vanilla fine-tuning always meets. However, TAIA can leverage out-of-domain data to further boost the performance of LLMs in out-of-domain scenarios, which could be useful under data-scarce scenarios.
> - Purpose of Figure 3: Figure 3 aims to show that with more exploration space (increasing tunable FFN layers), TAIA's attention parameters are sufficiently optimized and hence could leverage OOD data to improve over base models.
>
> Thank you for pointing out the confusion we have made and we will make it clear in the revised version.
> We are hoping to discuss our paper with you further!

---

> > ### Comment · Reviewer_VBCP · 2024-08-13
> >
> > Thanks for your response. I raise my score accordingly.

---

### Official Review · Reviewer_UWso · 2024-07-13

**Soundness:** 4
**Presentation:** 3
**Contribution:** 3
**Rating:** 7
**Confidence:** 3

**Summary:**

This paper proposes the hypotheses that under some circumstances LLMs are better trained by ft all parameters and then dropping the learnt FFNN layers. The proposed method is named TAIA from training all parameters but inferring with only Attention.

A comparison of TAIA's the performance together with another variants like TOA (train only attention) is carried out.
More over  other ablated version using the FF components instead of the attention were also evaluated.
 Several model are considered during the study such as Llama2 LLama3 and Qwen.

Several datasets (Alpaca , CoT-collection, ...) are used to train the models with the proposed technique and baselines.
Extensive comparison are also done with other OOD techniques such as L2, EC and self-distill, among others.
Several dimensions are taken into account such as varying data sizes.

The results are positive, showing the new proposed techniques improves performance when models are trained OOD.

Finally an honest limitation section with some experiments show conditions where the proposed method will not work better

**Strengths:**

The paper performs several experiments varying many factors to validate there method.

Several LLMS were used in the experiments, which helps into drawing generic conclusions.
Many datasets are considered for evaluating  2 aspects : reasoning and knowledge.
Ablation studies are carried out with other baselines varying TAIA is proposed.
Models are compared with other OOD techniques such as Self-distillation, L2, or WEC.
Representation analysis is carried out as well as quantitative analysis.

The discussions are very interesting with clear research questions for which corresponding experiments were executed.
Limitations include experiments to highlight some of the circumstances when the proposed technique doesn't work well.
Finally, the extensive appendices specify many details on experiment, additional results and detailed explanations that help the reading of the paper.

**Weaknesses:**

It would have nice to see what happens with even larger scale models.
In some of the experiments additional rounds or repetitions would have helped to understand the variability of the results.
Some deeper theoretical analysis could help to explain the unexpected and satisfactory results of the proposed technique.

**Questions:**

It is not very clear how the model is trained and used when we have in-domain and out-of-domain data. It looks like both are merged together loosing the differentiation. In such a case, do you think differentiation of the different sources would further improve the performance? maybe some additional experiments would be interesting. How would the FF updates be done in such a case in order to embed any missing in-domain knowledge into the model ?

**Limitations:**

Authors do an excellent work addressing and studying their proposal limitations.

---

> ### Author Rebuttal · Authors · 2024-08-04
>
> Thank you for your insightful reviews and comments! We are glad to hear that you find the proposed methodology useful and that it significantly improves over the previous OOD approaches. It is also encouraging to see that you find our experiments extensive, the analysis insightful and the explanations realistic.
>
> > TAIA in  Larger scale models.
>
> Response: Due to computational budgets, we conduct experiments based on the 14B and 32B sizes of Qwen using LoRA tuning and Alpaca-GPT4 data and show the results below:
>
> | Model  | Infer Mode | Reasoning |  |  |  |  | Knowledge  |  | Average |
> | -- | --- | --- | -- | -- | --- | --- | --- | -- | -- |
> |  |  | MATH  | BBH | CQA | LogiQA_EN | SVAMP | MMedBench en | MMLU  | |
> | Qwen1.5-7b  | Base  | 20.30 | 30.76 | 78.30 | 42.70 | 54.90 | 45.09  | 57.69 | 47.11 |
> | | LoRA  | 17.90  | 36.09 | 77.31 | 37.33 | 57.10 | 44.85  | 54.89 | 46.50   |
> |  | TAIA  | 24.98  | 43.46 | 77.31  | 41.78 | 67.20 | 46.90  | 57.29 | **51.27**   |
> | Qwen1.5-14b | Base  | 45.98  | 43.88 | 77.72 | 47.16 | 83.60 | 51.06 | 66.05 | 59.35   |
> |  | LoRA  | 38.74  | 41.65 | 74.45 | 41.78 | 82.80 | 48.15  | 63.71 | 55.90 |
> |  | TAIA  | 55.51  | 46.06 | 77.31| 46.39  | 83.10 | 52.55  | 65.48 | **60.92** |
> | Qwen1.5-32b | Base  | 41.22 | 48.78 | 80.92  | 50.23 | 87.60 | 61.04  | 73.03 | 63.26   |
> |  | LoRA | 39.12 | 46.29 | 77.81 | 49.31  | 82.60 | 59.54  | 71.02 | 60.81 |
> |  | TAIA | 42.70| 52.63 | 82.31 | 48.69 | 86.20 | 61.98  | 72.63 | **63.88**  |
>
> We observe that TAIA still outperforms both the base model and LoRA tuning model with Alpaca-GPT4 data, especially in reasoning tasks like MATH and BBH, which further verifies the effectiveness of TAIA.
>
> > Variability of TAIA
>
> Response: Thanks for your kind suggestions! We choose the Qwen1.5-7B model to conduct bias analysis by repeating three runs and present the results below to show the robustness of our method:
>
> | Model | Infer Mode | Reasoning |  |   |  | | Knowledge  |  | Average |
> | -- | --- | --- | --- | --- | --- | --- | --- | --- | -- |
> | | | MATH  | BBH   | CQA | LogiQA_EN | SVAMP | MMedBench en | MMLU | |
> | | Base | 20.30 | 30.76 | 78.30 | 42.70 | 54.90 | 45.09 | 57.69 | 47.11 |
> | Run1 | LoRA | 17.90 | 36.09 | 77.31 | 37.33 | 57.10 | 44.85 | 54.89 | 46.50 |
> | | TAIA  | 24.98 | 43.46 | 77.31 | 41.78 | 67.20 | 46.90 | 57.29 | 51.27 |
> | Run2  | LoRA   | 18.44  | 38.90 | 75.35  | 36.25 | 56.80 | 44.62  | 55.41 | 46.54   |
> | | TAIA   | 26.52 | 43.33 | 75.76 | 41.56 | 67.60 | 46.27 | 57.36 | 51.20 |
> | Run3  | LoRA  | 18.36 | 38.66 | 76.41| 36.87  | 56.70 | 44.46  | 55.39 | 46.69  |
> | | TAIA   | 27.02  | 43.56 | 76.41 | 41.56  | 67.90 | 46.11 | 57.12 | 51.38 |
>
> We observe that the improvements do not vary among different runs, which emphasizes the robustness of TAIA.
>
> > Theoretical explanation
>
> Response: We present the theoretical analysis of TAIA in our global rebuttal.
>
> > Differentiation of ID and OOD data
>
> Response: This is a great practice to see whether the differentiation of in-domain data and out-of-domain data will lead to further improvements. Following the experiment setting in Figure 1(c), we fine-tune Qwen1.5-1.8B models with LoRA where $r=16, \alpha=32$. We design three types of methods to differentiate the ID and OOD data during the fine-tuning process:
>
> - Experiment 1: Rather than mixing the ID and OOD data evenly in Figure 1(c), we adopt linear annealing to gradually decrease the proportion of ID data from 1 to 0 as training progresses.
>
> | Mixing Schedule  | Methods | OOD(20k) | OOD(40k) | OOD(60k) | OOD(80k) |
> | --- | --- | --- | --- | --- | --- |
> | Uniform  | Vanilla | 58.66 | 58.35 | 59.66 | 57.15 |
> |  | TAIA  | 64.74    | **64.59**  | 64.62  | 64.8  |
> | Linear Annealing | Vanilla | 60.42 | 58.9  | 60.95 | 63.16 |
> |  | TAIA  | **64.97** | 64.42  | **64.94**  | **65.18** |
>
> We found that the Linear Annealing method can mitigate the noise introduced by OOD data, while TAIA can further enhance the model's ability to utilize OOD data, thereby achieving higher performance on ID data.
>
> - Experiment 2: We fine-tune the Qwen1.5-1.8B with 20k ID data in the first stage and 20-80k OOD data in the second stage. Note that we adopt 2-stage fine-tuning based on TAIA-tuned models for better performance.
>
> |  | 1-stage | 2-stage  |  |  |   |
> | ---- | ---- | --- | --- | --- | --- |
> | Methods | ID(20k) | OOD(20k) | OOD(40k) | OOD(60k) | OOD(80k) |
> | Vanilla | 62.05   | 28.66  | 26.53  | 30.3  | 30.18  |
> | TAIA | **64.07**  | **49.71**  | **50.61**  | **47.43**  | **51.05**  |
> | TOA  | -   | 27.99 | 27.64 | 28.92   | 30.65  |
>
> The results show that the 2-stage fine-tuning of OOD data causes a serious performance drop. The first stage not only fits the model to the ID data distribution but also causes the model to lose its generalization ability, making it more sensitive to the noise introduced by OOD data. As a result, the 2-stage fine-tuning method severely degraded the model's performance.
>
> - Experiment 3: We follow experiment 2's setting but we fine-tune the Qwen1.5-1.8B with 20-80k OOD data in the first stage and 20k ID data in the second stage.
>
> |    | Methods | OOD(20k) | OOD(40k) | OOD(60k) | OOD(80k) |
> | --- | --- | --- | --- | --- | --- |
> | 1-stage   | TAIA    | **62.29**  | **61.85** | 61.03  | 61.23 |
> | 2-stage (w/ ID data) | Vanilla | 59.34  | 59.63  | 59.19 | 59.25  |
> |  | TAIA  | 60.68 | 59.6  | 60.97 | **61.44**  |
>
> Compared to the results of Experiment 2, the models fine-tuned on the OOD data in the 1-stage retain their generalization ability due to the data diversity of the CoT-Collection. However, the results show that such differentiation of the ID and OOD data still fails to improve performance further in the data mixing scenario.
>
> In summary, the mixing of ID and OOD data has already achieved similar performances with the linear annealing differentiation of both data, which prompts us to directly use the mixture as the training corpus.
>
> We are hoping to discuss with you further!

---

> ### Author Response · Authors · 2024-08-14
> **Thanks for valuable reviews**
>
> Dear reviewer:
>
> We hope this message finds you well.
>
> We kindly request your assistance in engaging in a discussion about our response. We have addressed your questions in our rebuttal, and we believe that a collaborative dialogue with you could be instrumental in ensuring a balanced and fair evaluation of our work.
>
> We are mindful of the demands on your time and the reviewers' time, and we greatly appreciate your consideration of this request.
>
> Thank you once again for your ongoing support and reviews throughout this process.

---

### Official Review · Reviewer_DXqC · 2024-07-13

**Soundness:** 3
**Presentation:** 3
**Contribution:** 3
**Rating:** 7
**Confidence:** 4

**Summary:**

This paper proposes a simple but effective strategy to improve fine-tuning of LLMs: fine-tune as normal but at inference time, only use the fine-tuned attention parameters but use the pre-trained MLP parameters. The intuition for this is that knowledge is generally stored in the MLP layers, and fine-tuning introduces harmful modifications to the pre-trained knowledge.

**Strengths:**

The results are strong, showing that TAIA improves over both vanilla fine-tuning and the baseline of only training attention, across a number of test datasets and models. It also improves over continual learning strategies, which have the same intuition of wanting to preserve aspects of the pre-trained model.

There is extensive analysis showing, among other results, that TAIA works both for LoRA and full fine-tuning, it is also better against red-teaming,

**Weaknesses:**

My main concern is from Figure 4a, which seems to show that the gap between Vanilla and TAIA decreases as the training dataset gets larger. This also made me realize the paper does not discuss how much training data is used in the main experiments. Figure 4 also does not disclose which model is being evaluated. I think it is important to address exactly how the results in Figure 4a compare with the CoT-Collection results in Table 1, where TAIA seems to be much better than Vanilla.

Relatedly, I would suggest discussing more about dataset size in Section 4.2.

**Questions:**

While I understand the intuition for TAIA, it also seems that it would introduce some distribution shift, as the fine-tuned attention parameters were trained to operate with the fine-tuned MLP parameters, but those are not present at inference time. Why does this not cause the model to perform badly?

Since you are doing fine-tuning, I wonder about the choice of using the chat models rather than base (no instruction tuning) models. Why use the chat models? Could you also do experiments on the base models?

Figure 3: What dataset is this accuracy reported on?

In Section 4.6, I wonder if you can also measure similarity to the pre-trained model. This would measure whether TAIA is essentially regularizing the model towards the pre-trained model (similar to what L2 or EWC would do, but apparently better).

I wonder if Section 5 should be called "Analysis" rather than "Discussion", as I usually think of Discussion as not introducing more results.

Table 4: Why is there a "-" in Infer Mode for the first row? I'm not sure what the first row represents.

Other comments:
* Line 22: Sentence should be rewritten as "which" currently attaches to NLP and not LLM

**Limitations:**

Limitations are appropriate.

---

> ### Author Rebuttal · Authors · 2024-08-03
>
> Thank you very much for your detailed reviews, comments, and suggestions. We are glad to hear that you find the TAIA a valuable method and that the experiments are sufficient to validate the method. Below we address the concerns and questions raised in the review.
>
> > Confusion about Section 4.2.
>
> Response: Sorry for the confusion we have made. We here address your concerns below:
> - Model used in Figure 4a: In terms of Figure 4a, we conduct experiments using Llama2-7b-chat with LoRA tuning.
> - Reasons for improvements decrease: The improvement decreases with the increasing data size since with the enlarged fine-tuning dataset, the distribution it covers is enlarged, which lessens the impact of the OOD data. We also notice that with relatively small sizes of data (100K-200K which is the common size of most datasets), TAIA consistently outperforms vanilla fine-tuning with non-negligible gains.
> - Dataset size of main experiments: The Alpaca-GPT4-bilingual contains 104K samples and the CoT-collection used in the main experiments is a 200K subset of a total 2M. The 200K subset is sampled from each category evenly to match with the scale of Alpaca-GPT4. We do not use full CoT-Collection dataset in main table due to limited computational resources.
>
> > Concern about the distribution shift
>
> Response: The removal of updated FFN parameters actually causes distribution shifts. However, in LoRA fine-tuning, what MLP has learned is quite limited [1], and hence causes little shift in the distribution learned in self-attention. In this case, the noise discarded brings further improvements than the side effects brought by the distribution shift. And we are also surprised to see that the removal of noise could bring such large improvements. When it comes to full-finetuning, where the distribution shifts are large, TAIA suffers from severe distribution shifts and performs inferior to the base model, while still outperforming LoRA methods.
>
> > Chat models instead of base models
>
> Response: We choose chat models based on the following reasons:
> 1. Fine-tuning based on the chat model is the commonly adopted practice to adapt a well-tuned model (with alignment tuning) to a downstream domain, including medical, or finance with minimal effort. This process often involves insufficient data that enables a base model to align well (which has also been claimed in our introduction) and we conduct our experiments based on this setting.
> 2. Base models are hard to evaluate: We find that base models cannot follow our evaluation instructions (like answer with more than the answer letter or the answer does not start with 'The answer is ') and result in severe true negative predictions. To reduce such bias, we use logit bias introduced by OpenAI which forces models to directly output the answer letter with additional logit scores. We conduct experiments on Commonsense QA, LogiQA en, MMLU and MMenBench en with a 1-shot manner and logit-bias technique to ensure that the evaluation results are convincing:
>
> | Model | Commonsense QA | LogiQA en | MMedBench en | MMLU  | Avg   |
> | ---- | ----- | ---- | ----- | ----- | ----- |
> | Llama2-7b-chat  | 48.40  | 33.95  | 32.21 | 42.30 | 39.22 |
> | Llama2-7b  | 27.93    | 20.28| 27.73  | 22.97 | 24.73 |
> | Llama2-7b-1shot | 29.40  | 21.20  | 32.29| 27.87 | 27.69 |
> | LoRA-1shot | 33.17  | 23.96  | 29.93| 28.26 | 28.83 |
> | TAIA-1shot  | 42.18 | 22.58 | 30.01  | 27.44 | 30.55 |
>
> We find that the performances of base models are significantly lower than chat models. Moreover, logit bias generation lacks the rationale generated in the CoT manner and hence still performs badly compared to chat models. Although TAIA still outperforms vanilla LoRA tuning with consistent gains, we still choose chat models as pre-trained models to maintain that models can follow instructions without demonstrations and ensure that each model can generate in a CoT style with higher performance.
>
> 3. TAIA is proposed for the OOD generalization of models. Using base models to train on the OOD dataset will never teach them to perform as expected in the downstream datasets, which may require an answer format different from that of training data, such as the answer option. In contrast, chat models are already instruction-tuned to follow diverse instructions, which corresponds to the motivation of TAIA.
>
> > Figure 3: What dataset is this accuracy reported on?
>
> Response: We report the accuracy on MMedbench, a medical reasoning evaluation set. We also include the detailed description of Figure 3 in Appendix E.6 in our original paper.
>
> > Similarity with pre-trained models of TAIA
>
> Response: Thanks for your suggestions. We present the similarity of TAIA and vanilla with the base model as below:
>
> | Model   | MATH   | Medical| CoT  |
> | -- | --- | ---- | ---- |
> |  | Sim/Acc | Sim/Acc | Sim/Acc |
> | Vanilla | 87.50/50.56 | 78.97/37.74 | 77.92/24.67 |
> | TAIF | 96.39/54.85 | 84.53/45.62 | 85.83/26.75 |
> | TAIA | 96.44/56.32 | 95.91/55.57 | 91.92/54.38 |
>
> where `Sim` represents the hidden state similarity with pre-trained model and `Acc` is the downstream performance on C-Eval. The results reflect that TAIA regularizes the fine-tuned model in an implicit manner without disturbing the learning and parameter exploration, which happens in other regularization methods like L2 and EWC.
>
> > Table 4: Why is there a "-" in Infer Mode for the first row? I'm not sure what the first row represents.
>
> Response: Sorry for the confusion. "-" means we directly use the base model without any fine-tuning. We will make this clear in the revised version.
>
> > Written concerns about the section name and grammar mistakes
>
> Response: Thanks for your kind suggestions. We will correct them in the revised version.
>
> We hope our responses could resolve your questions and we are hoping to discuss with you further!
>
> [1] Jiang T, Huang S, Luo S, et al. MoRA: High-Rank Updating for Parameter-Efficient Fine-Tuning[J]. arXiv preprint arXiv:2405.12130, 2024.

---

> > ### Comment · Reviewer_DXqC · 2024-08-12
> >
> > Thank you for your response! I believe this is a good paper and should be accepted.

---

### Official Review · Reviewer_y3YK · 2024-07-21

**Soundness:** 2
**Presentation:** 2
**Contribution:** 2
**Rating:** 4
**Confidence:** 4

**Summary:**

The paper proposes a novel inference-time intervention method that trains all model parameters but retains only the self-attention updates for inference. This approach, named "Training All parameters but Inferring with only Attention" (TAIA), optimizes performance across a variety of downstream and closed-book tasks, as validated by extensive experiments. The paper confirms the reproducibility of TAIA in terms of fine-tuning methods and dataset scales. TAIA also maintains the few-shot adaptation ability of base models and withstands multi-level adversarial attacks, consistently improving performance in vertical domains like healthcare even with increasing OOD data.

**Strengths:**

1. This paper introduced an innovative inference-time intervention strategy termed "Training All parameters but Inferring with only Attention" (TAIA), which trains all model parameters while retaining solely the refined self-attention updates for inference.
	2. Through extensive experimental validation on a variety of datasets and with LLMs of varying parameter sizes, the authors showcased that TAIA consistently outperforms both fully fine-tuned models and base models under most conditions, achieving substantial performance enhancement.
	3. The authors substantiate the robustness and generalization capabilities of TAIA across different fine-tuning methodologies and dataset scales. TAIA also preserves the base models' few-shot adaptation prowess and defends against multi-tiered adversarial assaults, markedly elevating performance in niche domains such as healthcare, even amidst escalating OOD data.

**Weaknesses:**

1. The authors posit that the parameters of Feed-Forward Networks (FFNs) encapsulate a substantial amount of pre-trained knowledge. However, these parameters are prone to shifts during Supervised Fine-Tuning (SFT), which may result in suboptimal performance on Out-Of-Distribution (OOD) data. But there is a notable absence of empirical evidence to substantiate this claim and merely citing other studies is insufficient.
2. The authors train the FFN parameters during the training process but discards them during inference. Could this lead to an inconsistency between the training and inference?
3. The authors lack implementation details on how to apply TAIA to the full finetune in response to RQ1.
4. The author should include a comparative experiment, utilizing only the parameters of the FFN while keeping the attention parameters fixed during the inference process. This will highlight the validity of TAIA.
5. The method section primarily contains text without any mathematical formulation, making it challenging to follow. Including formulations on how the model learns from different types of data and how it only infers with attention during inference would improve clarity and make it easier to evaluate the method's contribution.
6. Adding some theoretical proof to support your claims would strengthen the paper.
7. The overall results indicate a serious overfitting issue, as the base models have shown competitive results, such as 53.76 (base) vs. 53.24 (TAIA). Could you provide more insights into this observation?

**Questions:**

Refer to the above.

**Limitations:**

Yes

---

> ### Author Rebuttal · Authors · 2024-08-03
>
> Thank you for your review and important questions! We are glad to hear that the reviewer found our proposed TAIA innovative and robust. Below we address the concerns and questions raised in the review.
>
> > 1.Fine-tuning of FFN leads to distribution shift
>
> Response: To empirically show the distribution shifts of FFN after fine-tuning, we follow Hu et al. [1] to conduct a forgetting analysis in the WikiText-103 test dataset. This dataset has already been used for LLMs' pretraining and we can compare the cross-entropy loss of the base model (as an anchor), vanilla LoRA tuning and TAIA to see the distribution shifts introduced by FFN module. A higher loss value indicates that the alteration of parameters brings distribution shifts that result in suboptimal performance on OOD data, since LLMs' generalization ability relies heavily on pre-trained knowledge. We choose Llama3-8b to serve as the base model and use CoT-collection, Alpaca-GPT4 and a more complex dataset, Tulu v2 as the training data.
>
> | Method | CoT-Collection | Alpaca-GPT4 | Tulu v2 |
> | ------ | -------------- | ----------- | ------- |
> | Base   | 3.7016         | 3.7016      | 3.7016  |
> | LoRA   | 6.0012         | 3.7396      | 3.7102  |
> | TAIA   | 5.9295         | **3.6934**      | **3.5545**  |
>
> We can see that with the increasing quality of the dataset (CoT-Collection->Alpaca-GPT4->Tulu v2), vanilla LoRA tuning suffers from lower distribution shifts, but still has a higher loss than the base model, which exemplifies that in OOD tuning scenario, FFN will cause catastrophic forgetting and distribution shifts. In contrast, TAIA which removes the shifted FFN parameters, results in even lower loss compared to base when choosing appropriate training datasets (Alpaca-GPT4 and Tulu v2). This also indicates that self-attention learns only the beneficial knowledge to augment the pre-trained knowledge but introduces little distribution shifts in the internal distribution of LLMs.
>
> > 2. Does TAIA lead to an inconsistency between the training and inference?
>
> Response: The removal of updated FFN parameters actually causes distribution shifts. However, in LoRA fine-tuning, what MLP has learned is quite limited [1], and hence causes little shift in the distribution learned in self-attention. In this case, the discarded disrupting knowledge brings further improvements than the side effects brought by the distribution shift. When it comes to full fine-tuning, where the distribution shifts are large, TAIA suffers from severe inconsistency between training and inference and performs inferior to the base model, while still outperforming LoRA methods.
>
> > 3 and 5. Implementation of TAIA
>
> Our response: During training, the parameters of FFN and self-attention are updated based on max-likelihood modeling:
>
> $$
> \{\theta\}\_{ffn}',\{\theta\}\_{attn}'=\arg\max_{\\{\theta\\}} \sum_{i=1}^N p(\boldsymbol{y}\_i|\boldsymbol{X}\_i,\{\theta\}\_{ffn},\{\theta\}\_{attn})
> $$
>
> where $N$ is the number of training samples and $\boldsymbol{X}\_i,\boldsymbol{y}\_i$ are query and response sequences sampled from any conversational data, like Alpaca-GPT4 or CoT-Collection, respectively. $\{\theta\}'\_{(\cdot)}$ is the updated weight in full fine-tuning or the merged weight of LoRA tuning.
> After training, TAIA only utilizes the updated attention parameters and reuses the pre-trained FFN parameters to perform inference:
>
> $$
> \boldsymbol{y}=\arg\max_{\boldsymbol{y}} \sum\_{j=1}^K \log p(y\_j|\boldsymbol{y}\_{j-1}, \boldsymbol{X}, \{\theta\}\_{ffn}, \{\theta\}\_{attn}')
> $$
>
> where $K$ is the generated sequence length and $\boldsymbol{X}$ is the query input to LLMs that shares different distributions with the training data. In this scenario, $\{\theta\}\_{ffn}$ is the original parameter of FFN in pre-trained models and $\{\theta\}\_{attn}'$ is the updated parameter groups of self-attention in full fine-tuning (or merged weight of self-attention in LoRA tuning).
>
> > 4. The validity of TAIA.
>
> Response: We have conducted experiments to see if we utilize only the parameters of FFN modules while keeping the attention parameters fixed during inference (which we call TAIF) in Table 5 in Appendix.
>
> > 6. Adding some theoretical proof to support your claims would strengthen the paper.
>
> Response: We offer a possible explanation of TAIA in the global rebuttal.
>
> > 7. The overall results indicate a serious overfitting issue, as the base models have shown competitive results, such as 53.76 (base) vs. 53.24 (TAIA). Could you provide more insights into this observation?
>
> Response: This is a phenomenon when the training data is insufficient for the base model and out-of-distribution against the downstream evaluation sets. For example, the CoT-collection derives from FLAN-v2, which is quite outdated and has lower complexity and diversity than Alpaca-GPT4 which is generated using a GPT4. Meanwhile, the base model, Llama3-8b, is relatively strong, so the data quality of CoT-Collection will never benefit the base model in such OOD settings. Under such a scenario, the vanilla fine-tuning method brings severe catastrophic forgetting problems while our TAIA can achieve competitive results with the base model. Once utilizing a relatively high-quality dataset like Alpaca-GPT4, TAIA can still outperform the base model, such as 54.54 (TAIA) vs. 53.76 (base) under out-of-distribution fine-tuning scenarios.
>
> We are glad to discuss with you further!

---

> ### Comment · Reviewer_y3YK · 2024-08-13
>
> Thank you for the response. I have updated my score from 3 to 4 as the authors addressed some of my concerns. I still think the proposed method leads to an inconsistency issue between the training and inference stages (a common concern raised by reviewers). The authors are encouraged to further investigate how to alleviate this issue with more method justification.

---

> > ### Author Response · Authors · 2024-08-13
> > **Thanks for your reply**
> >
> > Thanks a lot for your comments! We are glad to see that you increased the score to 4. Below we further clarify the concerns raised in the comment.
> >
> > We are focusing on adapting LLM to specific downstream tasks via OOD training corpus (which is easy to retrieve). Under such circumstances, the training and inference are naturally inconsistent where the training is conducted under one data distribution, but the inference is performed under another distinct data distribution. Such inconsistency derives from the shifted pre-trained knowledge stored in the FFN module during fine-tuning on an OOD dataset (see the first table in our response to you). Therefore, TAIA intuitively removes such inconsistency by reusing the pre-trained FFN parameters while adopting the updated self-attention modules. The updated self-attention module brings little distribution shift to the base model and enhances the reasoning ability of LLMs. That is why TAIA can improve over the baselines instead of degrading performance.
> >
> > In our response to Reviewer UWso and Figure 1 of our original paper, we present the severe inconsistency brought by OOD fine-tuning, and that TAIA can mitigate such distribution inconsistency and leverage the only beneficial parts of fine-tuning corpus to improve over pre-trained models.
> >
> > On top of the TAIA method, we want to also highlight our contributions to the investigation on the roles of FFN and self-attention module during fine-tuning and extensive analysis that helps us better understand how LLMs benefit from the TAIA method when only provided with OOD data.
> >
> > Thanks again for all your time and efforts in giving us insightful comments.

---

### Author Rebuttal · Authors · 2024-08-05

We offer a possible explanation of TAIA.

Suppose an LLM $p_{\theta}$ containing pretrained weight $\theta_0$, the vanilla LoRA-tuned model yields $\Delta \theta_0$ weight which is to be merged back to pretrained weight and has a relatively small norm. Suppose a simplified neural network layer with nonlinear operators, a layer normalization layer $\mathrm{LayerNorm}(X)=\frac{X-\mu}{\sigma}$ and a residual connection:

$$
M_{W}(X)= \text{Act}(Wx)
$$

As the magnitude of $\Delta \theta_0$ is quite small compared with $\theta_0$, we perform a first-order Taylor expansion on $M_{\theta_0+\Delta \theta_0}(X)$:

$$
M_{\theta_0+\Delta \theta_0}(X)\approx M_{\theta_0}(X)+J_{\theta_0}(X)\Delta \theta_0
$$

where $J_{\theta_0}(X)$ is the Jacobian matrix of $M_{\theta_0}(X)$ for $\theta_0$.
We define $z=X+M_{\theta_0}(X)$ and $z'=X+M_{\theta_0}(X)+J_{\theta_0}(X)\Delta \theta_0$ to compare $\mathrm{LayerNorm(z)}$ and $\text{LayerNorm}(z')$. Compare the addition of $\Delta \theta_0$ inside the transformer module:

$$
X=\text{LayerNorm}(z) \quad X'=\text{LayerNorm}(z')
$$

Considering $z'=z+J_{\theta_0}(X)\Delta \theta_0$, we have

$$
\mu_{z'}\approx \mu_z+\mu_{J_{\theta_0}(X)\Delta \theta_0} \quad
\sigma_{z'}\approx \sigma_z
$$

Since $\|\Delta \theta_0\|$ is quite small compared to $\theta_0$, $J_{\theta_0}(X)\Delta \theta_0$ is also of small norm, which can be considered to be ignored for the mean and standard deviation. Based on it, we have

$$
\text{LayerNorm}(z')\approx \text{LayerNorm}(z)
$$

In other words, if the updated parameter is small enough compared to the pretrained model, the output distribution after each submodule remains consistent with the pretrained model. Meanwhile, we need to prove that the perturbation brought by attention is far lower than that brought by FFN so that the removal of FFN results in higher improvement than the removal of attention modules. Specifically, omitting the multi-head mechanism, self-attention is formalized as such:

$$
\text{Attn}(X) = \text{SoftMax}\left(\frac{XW_q (XW_k)^\top}{\sqrt{d_k}}\right)XW_v
$$

where $X \in \mathbb{R}^{n \times d}$, and $W_q, W_k\in\mathbb{R}^{d \times d_k}, W_v\in\mathbb{R}^{d \times d_v}$, respectively.

For small perturbations $\Delta W_q, \Delta W_k, \Delta W_v$, the perturbed Attention output is:

$$
\text{Attn}_\Delta(X) = \text{SoftMax}\left(\frac{X(W_q + \Delta W_q) (X(W_k + \Delta W_k))^\top}{\sqrt{d_k}}\right)X(W_v + \Delta W_v)
$$

Using the first-order Taylor expansion, we have:

$$
\Delta \text{Attn}(X) \approx \frac{\partial \text{Attn}(X)}{\partial W_q} \Delta W_q + \frac{\partial \text{Attn}(X)}{\partial W_k} \Delta W_k + \frac{\partial \text{Attn}(X)}{\partial W_v} \Delta W_v
$$

Using the first-order Taylor expansion and the partial derivatives of softmax, we obtain the perturbation bound of attention:

$$
\\|\Delta \text{Attn}(X)\\| \leq \frac{1}{\sqrt{d_k}} (\\|(\text{diag}(P)-PP^{\\top})X(XW_k)^{\\top}XW_v \\|\\|\Delta W_q\\| + \\|(\text{diag}(P)-PP^{\\top})XW_q^{\\top}XXW_v \\|\\|\Delta W_k\\|) + \\|P^{\\top}X\\|\\|\Delta W_v\\|
$$
where $P=\text{SoftMax}(\frac{XW_q(XW_k)^{\top}}{\sqrt{d_k}})$
We simplify the bound as such  $\\|\Delta \text{Attn}(X)\\| \leq C_{attn}(\\|\Delta W_q\\|+\\|\Delta W_k\\|+\\|\Delta W_v\\|)$ where $C_{attn}=\frac{\\|X\\|^3\\|W_k\\|\\|W_v\\|}{\sqrt{d_k}}\\|\text{diag}(P)-PP^{\\top}\\|$.

In terms of FFN modules, we use ReLU as the activation function instead of SwiGLU in FFN, which is defined as:

$$
\text{FFN}(X) = \text{ReLU}(W_1(\text{ReLU}(W_2x + b_2)) + b_1)
$$

where $W_1 \in \mathbb{R}^{d \times 4d}$, $W_2 \in \mathbb{R}^{4d \times d}$.

For small perturbations $\Delta W_1, \Delta W_2, \Delta b_1, \Delta b_2$, the perturbed FFN output is:

$$
\text{FFN}_\Delta(X) = \text{ReLU}((W_1 + \Delta W_1)(\text{ReLU}((W_2 + \Delta W_2)X + b_2 + \Delta b_2)) + b_1 + \Delta b_1)
$$

Similar to the above analysis, we obtain the perturbation bound of FFN modules:

$$
\\|\Delta \text{FFN}(X)\\| \leq \\|f'(Y)XW_1f'(XW_2+b_2)\\|\\|\Delta W_2\\|+\\|f'(Y)W_1f'(X_2+b_2)\\|\\|\Delta b_2\\| + \\|f'(Y)Y\\|\\|\Delta W_1\\| + \\|f'(Y)\\|\\|\Delta b_1\\|
$$

where $f'$ is the derivative of $\text{ReLU}$ and $Y=\text{ReLU}(XW_2+b_2)$ and we simplify the bound as such: $\\|\Delta \text{FFN}(X)\\| \leq C_{ffn}(\\|\Delta W_2\\|+\\|\Delta W_1\\|+\\|\Delta b_1\\|+\\|\Delta b_2\\|)$ where $C_{ffn}=nd\\|X\\|\cdot\\|W_1\|b_1\\|\cdot\\|W_2|b_2\\|$.

The SoftMax function’s normalization limits the perturbation’s amplification and hence limits the $\\|\text{diag}(P)-PP^{\\top}\\| \leq 1$ and the overall perturbation. Besides, $\\|X\\|_2^2\sim\chi^2, \\|X\\| \ \approx \sqrt{d}$. Due to ReLU’s non-linear activation, perturbations are significantly amplified around the non-linear activations, especially in the high-dimensional space. Based on the above comparison

$$
C_{attn}\approx d\\|W_k\\| \\|W_v\\| \\|\text{diag}(P)-PP^{\\top}\\| \ll C_{ffn}=nd\sqrt{d}\cdot\\|W_1\|b_1\\|\cdot\\|W_2|b_2\\|
$$

which implies

$$
\\|\Delta \text{Attn}(X)\\| < \\|\Delta \text{FFN}(X)\\|
$$

Thus, for equal-magnitude parameter perturbations, the output perturbation bound of the Attention module is less than that of the FFN module. This indicates that the FFN updates will introduce great perturbation to the distribution of transformers and narrow the similarity between tuned models and pre-trained models. **In fact, TAIA keeps the most similar delta parameters with the pretrained model and thus achieves the best performance among all the methods. The similarity comparison between TAIA and the base model is presented in the rebuttal to Reviewer DXqC.**

---

> ### Comment · Reviewer_DXqC · 2024-08-12
>
> This is interesting, although I don't think it explains why TAIA outperforms freezing the FFN and only training attention (TOA).
>
> (That said, I think the paper's empirical results are strong enough on their own to warrant acceptance)

---

### Author Response · Authors · 2024-08-12
**Thanks for valuable reviews**

We hope this message finds you well.

First and foremost, we would like to express our sincere gratitude for your efforts in overseeing the review process. We deeply appreciate the time and attention that you have devoted to evaluating my submission.

We are writing to kindly inquire if it would be possible to engage in the discussion. We have responded to your questions in our rebuttal and we believe that a collaborative discussion with you could be highly beneficial in ensuring a balanced and fair assessment of our work.

We understand the demands on your time, and we truly appreciate your consideration of this request. We are hoping to receive your thoughtful feedback.

Thank you once again for your continued support and valuable reviews about our work.

---

### Decision · Program_Chairs · 2024-09-25

**Decision:**

Accept (poster)

**Comment:**

This paper introduces an interesting fine-tuning method for LLMs, TAIA (Training All parameters but Inferring with only Attention). It aims to enhance the performance of large language models (LLMs) in specialized, data-scarce domains where training data is not fully aligned with the downstream task distribution. It claims that FFN holds more pretrained knowledge while Attention holds more flexible, domain-related knowledge. As a result, during inference, the method retrains only the fine-tuned attention parameters, while discarding modifications to feed-forward networks (FFN). The authors provide experimental validation showing the effectiveness of TAIA across various datasets and model sizes.

Main concerns with the paper center around the theoretical grounding. While the proposed method, TAIA, shows promising empirical results, there is a lack of a detailed theoretical analysis to support the intuition. And I also agree that the claim about that FFN holds more pretrained knowledge is a strong claim, and the paper lack sufficient analysis, I would suggest the authors to weaken the expressions.

Another concern is the potential inconsistency introduced between the training and inference stages. While TAIA suggests that discarding FFN parameters during inference is beneficial, this approach raises questions about the distributional shift that may occur.

Despite these concerns, the paper presents a novel and potentially impactful idea that could influence research in fine-tuning LLMs. The experiments are solid enough.